# Perioperative pupil size in low-energy femtosecond laser-assisted cataract surgery

**Alireza Mirshahi**[1]*, **Astrid Schneider**[2], **Catharina Latz**[1], **Katharina A. Ponto**[3,4]

**1** Dardenne Eye Hospital, Bonn, Germany, **2** Institute for Medical Biostatistics, Epidemiology and Informatics, University Medical Center Mainz, Mainz, Germany, **3** Department of Ophthalmology, University Medical Center Mainz, Mainz, Germany, **4** Center for Thrombosis and Hemostasis, University Medical Center Mainz, Mainz, Germany

* Mirshahi@dardenne.de

## Abstract

### Purpose

To assess potential changes in pupil size during femtosecond laser-assisted cataract surgery (FLACS) using a low-energy laser system.

### Methods

The pupil sizes of eyes undergoing FLACS were measured using the Ziemer LDV Z8 by extracting images from the laser software after each of the following steps: application of suction, lens fragmentation, and capsulotomy. Furthermore, the pupil diameters were measured based on preoperative surgical microscope images and after releasing the suction. Paired *t*-test and the two one-sided tests (TOST) procedure were used for statistical analyses. The horizontal and vertical pupil diameters were compared in each of the steps with preoperative values.

### Results

Data were available for 52 eyes (52 patients, mean age 73.4 years, range 51–87 years). The equivalence between mean preoperative pupil size and status immediately after femtosecond laser treatment was confirmed ($p<0.001$; 95% confidence interval [−0.0637, 0.0287] for horizontal and $p<0.001$; 95% CI [−0.0158, 0.0859] for vertical diameter). There was statistically significant horizontal and vertical enlargement of pupil diameters between 0.15 and 0.24 mm during the laser treatment steps as compared with preoperative values (all *p* values <0.001).

### Conclusions

No progressive pupil narrowing was observed using low-energy FLACS. Although a suction-induced, slight increase in pupil area became apparent, this effect was completely reversible after removing the laser interface.

**Data Availability Statement:** All relevant data are within the manuscript and its Supporting information files.

**Funding:** K.A.P. is funded by the German Federal Ministry of Education and Research (BMBF 01EO1003).

**Competing interests:** AM is a consultant to and has received honoraria for speaking at meetings by Ziemer Ophthalmics, Biel, Switzerland. Ziemer assisted in extraction of picures from the femtosecond laser device. This does not alter our adherence to PLOS ONE policies on sharing data and materials.

## Introduction

During the past decade, femtosecond laser-assisted cataract surgery (FLACS) has gained increasing popularity [1–4]. However, its automation and precision come at a price [5,6]. One of these costs is miosis induction. This phenomenon has been described in several studies using different laser platforms [7,8] and is believed to be induced by increased levels of prosta-glandin $E_2$ ($PGE_2$) measured at the end of laser pretreatment [7–10]. Currently, preoperative administration of nonsteroidal anti-inflammatory drugs (NSAIDs) is recommended to overcome the alteration of pupil size [9,10]. In their assessment of the Catalys Precision Laser System, Jun et al. reported a correlation between the degree of miosis and patient age, time for lens fragmentation, and time for main incision creation [8]. However, according to our personal experience in the clinic we hypothesized that no miosis induction through laser pretreatment occurs when using the Ziemer Z8 laser. This could be because of the Ziemer Z8 inherent low-energy–high-frequency laser technology. Conventional "high-energy" femtosecond lasers emit pulses with an energy in the 4 to 15 μJ range [5,7,11], whereas the newer low-energy technology uses high-pulse repetition rates greater than 1 MHz and a low-pulse energy in the nanojoule range [12]. This is achieved using a high numerical aperture in the laser-focusing optics, enabling small laser spot sizes [13].

This study aims to better understand the possible factors influencing changes in pupil size in FLACS, by analyzing pupil size at different surgical time points.

## Materials and methods

Fifty-two eyes of 52 patients undergoing routine FLACS were included in this retrospective study. The patients' age, laterality of surgery, and special notes in the surgery report were extracted for analysis as well as all known ocular comorbidities such as pseudoexfoliation syndrome and glaucoma. Axial length, anterior chamber depth, lens thickness, and white-to-white distance were obtained from laser biometry performed on the day of surgery (IOLMaster 700, Carl Zeiss Meditec). If both eyes were eligible for the study, only the data from the first operated eye were used. This study was performed according to the tenets of the Declaration of Helsinki and did not require approval of an independent ethics committee, as ruled by the North Rhine Medical Chamber.

### Pupil size

Retrospectively, the images of the top-view camera that were integrated into the laser handpiece and taken automatically during the liquid interface docking of the eye were analyzed with regard to the horizontal and vertical pupil diameters at the following surgical time points: immediately after application of the vacuum docking suction, after completion of lens fragmentation, and after completion of laser capsulotomy. In addition, the diameters of the pupils before surgery and after release of suction and removal of the laser handpiece were measured by analyzing the images from the video recordings taken through the surgical microscope.

**Normalization of pupil diameter raw data for cornea magnification in different media.**   When the pupil is observed under a surgical microscope by looking through the cornea, the refractive power of the cornea magnifies the pupil, unlike objects outside the cornea, such as the limbus. Although the laser handpiece is docked to the eye, the space above the cornea is filled with a balanced salt solution (BSS) instead of air, as with microscopy. This results in a reduction in the refractive power of the cornea–BSS interface as compared to the cornea–air interface due to the change in the index of the refraction difference (Fig 1).

The resulting change in magnification was modeled using ZEMAX ray-tracing software and the Liou-Brennan eye model [14]. A correction factor to compensate for the reduced pupil

a) Microscope observation

b) Observation through BSS during laser docking

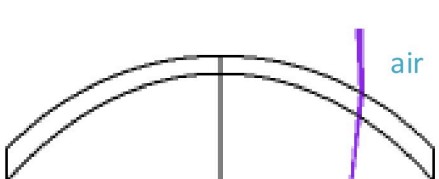

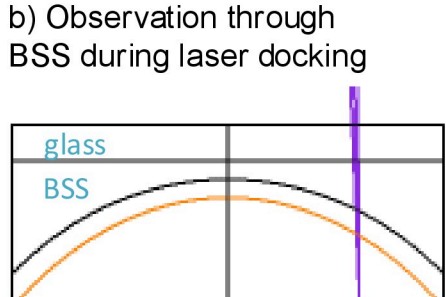

**Fig 1. Illustration of the difference in magnification when observing the pupil through the cornea (a) by microscope through the air–cornea interface versus (b) during liquid interface laser docking.**

magnification in the images taken by the laser was determined and applied to the measured raw pupil diameters.

## Surgical technique

One single experienced cataract surgeon (A.M.) performed all FLACS procedures using the Ziemer LDV Z8 (Ziemer Ophthalmic System) and an Alcon Centurion phacoemulsification system (Alcon Lab.). All surgeries were performed at the Dardenne Eye Hospital in Bonn, Germany.

Routine preoperative care included mydriatic treatment consisting of tropicamide 5 mg/mL (Mydriaticum Stulln® UD, Pharma Stulln GmbH) and phenylephrine 5% (Neosynephrin-POS® 5%, Ursapharm Arzneimittel GmbH) eye drops, four times each. NSAIDs were not administered to any patient. All surgeries were performed under peribulbar anesthesia (each 3 ml of Mepivacaine 2% and Bupivacaine 0.75% and 75 IE hyaluronidase with a 22 Gauge needle) with the surgeon sitting at the 12 o'clock position and the Ziemer Femto LDV Z8 positioned at an oblique angle. After disinfection and sterile draping, the femtosecond laser interface was docked, and vacuum suction was applied (approximately 420 mbar). Standard laser parameters were a 6 mm diameter laser lens fragmentation in six segments at 105% laser energy, followed by a 5.2 mm capsulotomy diameter using 90% laser energy. Suction was released after the completion of capsulotomy, and the laser was moved aside. All subsequent surgical steps were performed under a surgical microscope (Zeiss). A posterior limbal main incision of 2.8 mm width and two 1.1 mm paracenteses were made. Dispersive viscoelastic was introduced into the anterior chamber. The laser-precut capsulotomy was removed using Utrata forceps. The lens was hydrodissected from the capsule, and the nucleus was hydrodelineated. The nucleus was removed using high-vacuum phacoemulsification. Cortex aspiration, polishing of the posterior capsule, and removal of the viscoelastic device after implantation of the intraocular lens were performed using a bimanual irrigation-aspiration system. At the end of surgery, the paracenteses were hydrated and controlled for water tightness.

## Statistical analyses

Normally distributed continuous variables are described using mean ± standard deviation (SD), minimum, and maximum. Skewed variables are described using median, 25%, and 75% quartile. Boxplots of the individual values as well as the difference from the first value (before surgery) are presented. Categorical variables are described using relative and absolute frequencies.

The main question to address was whether the measures "before surgery" and "released suction," each measured on the same subject, were equivalent; that is, whether they differed on average by a small margin. A difference of >|0.2 mm| was assumed to be clinically relevant. Thus, an equivalence range of [−0.2, 0.2] was fixed. To test a possible equivalence between two measurements, the two one-sided tests (TOST) method was carried out, according to the work by Schuirmann [15,16]. For this, TOST were determined to reject the null hypotheses mu < 0.2 and mu > −0.2. If both null hypotheses can be rejected, the equivalence to the given range is proven. In addition, the 90% confidence interval of the difference is given. These were confirmatory analyses with a global significance level of 5%. To determine how the individual differences behave compared with the pupil diameter, a mean difference plot was also generated.

A secondary question was whether the values measured during different steps of the laser pretreatment varied from the initial values. Whether the effect was clinically relevant or whether equivalence could still be assumed were investigated. Two-sided paired *t*-tests were performed to investigate these differences. In addition, a possible equivalence with an equivalence range [−0.2, 0.2] was investigated using the TOST method. No significance-level adjustments were made for the secondary questions. The *p* values were used for descriptive purposes. The statistical evaluation was carried out using the software program R, version 3.5.3 (http://www.R-project.org).

## Results

Complete data were available for 52 eyes of 52 patients (mean age 73.4 years, range 51–87 years; 19 right and 33 left eyes). Glaucoma was diagnosed preoperatively in four eyes (7.7%), and pseudoexfoliation was diagnosed in five eyes (9.6%). Further descriptive data, including axial length, anterior chamber depth, lens thickness, and white-to-white distance, are illustrated in Table 1.

The mean (±SD), median [minimum-maximum] preoperative values were 7.06 (±0.66), 7.12 [5.5–8.27 mm], respectively, for horizontal pupil diameter and 7.16 (±0.65) mm, 7.12 [5.75–8.46 mm], respectively, for vertical pupil diameters. The changes in pupil diameters at the different time stages are shown in Table 2 and Fig 2.

The horizontal and vertical pupil diameters were equivalent when comparing preoperative measurement to post laser values; the mean horizontal difference between these values was −0.0175 mm. The TOST procedure confirmed that the difference was within the preset

**Table 1. Preoperative morphologic parameters in 52 eyes undergoing low-energy Femtosecond-Laser-Assisted Cataract Surgery (FLACS).**

| Preoperative parameter | Mean ± standard deviation (mm) | Median (mm) | Minimum (mm) | Maximum (mm) |
|---|---|---|---|---|
| Axial length | 24.06 ± 1.35 | 24.10 | 21.34 | 27.13 |
| Anterior chamber depth | 3.23 ± 0.4 | 3.26 | 2.39 | 4.01 |
| Lens thickness | 4.62 ± 0.4 | 4.67 | 3.69 | 5.34 |
| White-to-white distance | 11.99 ± 0.46 | 12.0 | 10.8 | 12.8 |

**Table 2. Intraoperative pupil diameters at different points in time (mean ± standard deviation).**

| | Before surgery | Suction applied | Lens fragmentation completed | Capsulotomy completed | Suction released (laser completed) |
|---|---|---|---|---|---|
| Horizontal diameter (mm) | 7.06 ± 0.66 | 7.20 ± 0.72 | 7.21 ± 0.71 | 7.24 ± 0.71 | 7.07 ± 0.65 |
| Vertical diameter (mm) | 7.16 ± 0.65 | 7.38 ± 0.71 | 7.40 ± 0.70 | 7.39 ± 0.71 | 7.13 ± 0.63 |

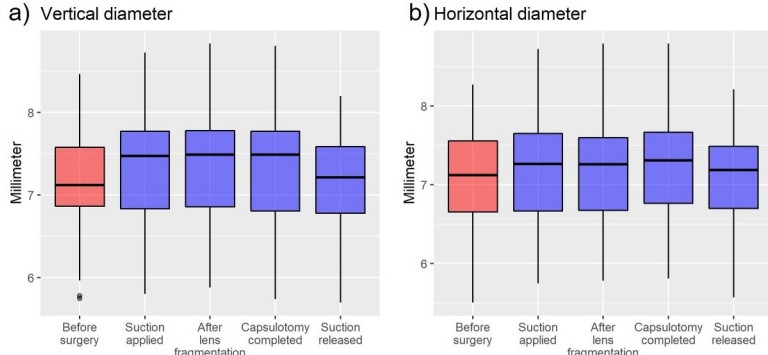

**Fig 2. Changes in pupil diameters at different time points of low-energy Femtosecond-Laser-Assisted Cataract Surgery (FLACS) using the Ziemer LDV Z8.** Boxplots showing the vertical (a) and horizontal (b) pupil diameters (median, quartiles, minimum, and maximum) before surgery, immediately after application of suction, after completion of lens fragmentation, after completion of laser capsulotomy, and after release of suction.

equivalent range of 0.2 mm ($p<0.0001$, 95% TOST interval [−0.0637, 0.0287]). The mean vertical difference between the stated values was 0.0351 mm. The TOST procedure confirmed that the difference was within the preset equivalent range of 0.2 mm ($p<0.0001$, 95% TOST interval [−0.0158, 0.08595]). Mean difference plots show that the mean and 95% TOST intervals were within the equivalence range (Fig 3).

We could not detect any equivalence of the preoperative pupil diameters to those measured immediately after application of suction, after completion of lens fragmentation, or after completion of laser capsulotomy (suction applied during those measurements, TOST procedure, all $p>0.05$). As shown in detail in Table 3, there was a statistically significant enlargement of pupil diameters (between 0.15 and 0.24 mm) during laser treatment steps compared with preoperative values (paired $t$-test, all $p$ values $< 0.001$).

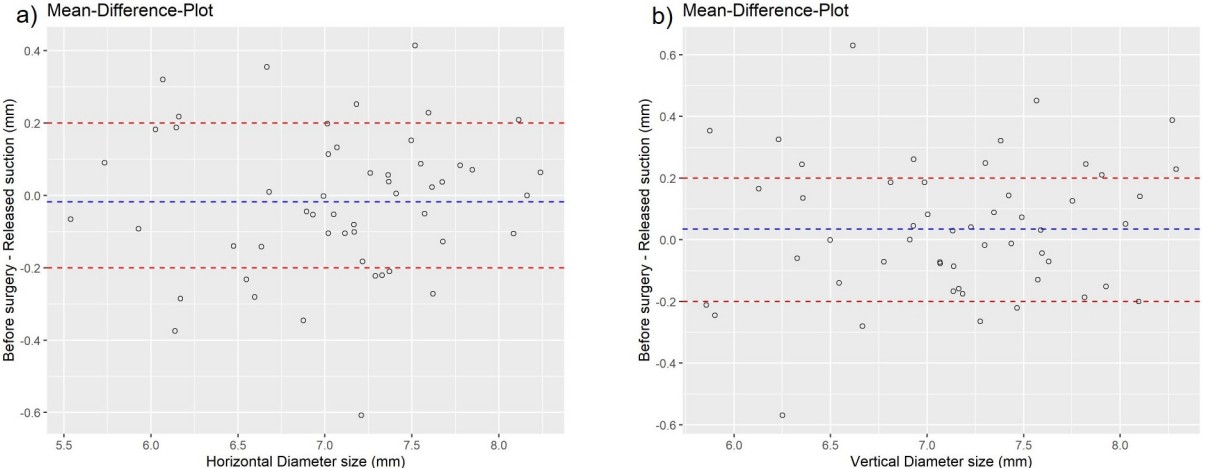

**Fig 3. Differences in the measures of the horizontal diameter (a) and vertical diameter (b) of the pupil before surgery and at the time point "released suction," each measured on the same subject.** The mean differences and the 95% confidence intervals of the differences are within the equivalence range [−0.2, 0.2].

**Table 3. Statistical analyses of change in pupil diameter within laser application steps as compared with preoperatively (paired *t*-test; mean change, *p* value [95% confidence interval]).** A negative difference indicates increase in size as compared with the preoperative size.

| Compared point in time | Suction applied | Lens fragmentation completed | Capsulotomy completed |
|---|---|---|---|
| Horizontal diameter (mm) | −0.145, $p < 0.001$, [−0.217, −0.073] | −0.152, $p < 0.001$, [−0.228, −0.076] | −0.183, $p < 0.001$, [−0.259, −0.108] |
| Vertical diameter (mm) | −0.222, $p < 0.001$, [−0.305, −0.139] | −0.238, $p < 0.001$, [−0.320, −0.155] | −0.225, $p < 0.001$, [−0.306, −0.145] |

## Discussion

To the best of our knowledge, this is the first study to measure pupil diameters at various points of low-energy FLACS. The main finding is this research is that overall, no change in the size of the pupil dimensions was noted from before the operation to after laser treatment, and a reversible and mild increase in the pupil area was noted during the laser procedure itself. This is in contrast to various studies that found FLACS to be associated with pupillary constriction [5,8,17–22]. In a previous study on low-energy FLACS with a smaller sample size, there was no statistically significant difference between the pupil areas measured preoperatively versus after laser treatment in the same surgical setting [23]. The current study confirmed these findings and furthermore analyzed the variations in pupil size during femtosecond laser application and under suction. The pupil size increased slightly under suction and then decreased to the preoperative level after suction was terminated. This is a physiologic phenomenon which is explained both by the afferent pupillary defect and also by an impairment in the iris sphincter (due to an intraocular pressure elevation greater than the systolic ophthalmic artery pressure) [24].

Femtosecond laser-induced miosis has been investigated in various studies [5–8,10,17–22]. In particular, each study has attempted to elucidate causative factors. Prostaglandin release has been discussed as a causative agent for miosis, and increased amounts have been measured in FLACS compared to classic phacoemulsification cataract surgery [18,20,21,25–27]. This was true for both low-energy, high-frequency as well as conventional femtosecond laser technology [8,21,27,28], although prostaglandin levels measured in low-energy, high-frequency FLACS were lower than in conventional FLACS. Consequently, pretreatment with NSAIDs before FLACS has been promoted and shown to reduce the prostaglandin surge, thereby decreasing FLACS-associated miosis [5,9,10,17,18,22,28]. Nevertheless, NSAID pretreatment does not appear to completely eliminate FLACS-induced miosis [18]. Therefore, additional causative agents were subsequently sought, and even technical details were examined to better understand this phenomenon. Bali et al. described a learning curve in his early experience with FLACS and found a miosis rate of 24% in his first 50 cases as opposed to 9.5% in the following 150 cases [11]. Suction time and docking attempts were significantly higher in the first 50 cases and were therefore postulated as causative factors of miosis. Similar results were reported by Roberts et al. when comparing the first 200 and subsequent 1300 cases [6]. In contrast, Jun found a 29% reduction in pupil area in 56 eyes undergoing high-energy FLACS [8]. This reduction was correlated with laser treatment duration, primary incision, and patient age but not with suction and shifting time (i.e., the time between the completion of femtosecond pretreatment and initiation of phacoemulsification). Although Schultz et al. [20] found no correlation between suction time and increased prostaglandin levels, Popiela et al. [19] reported significant miosis when suction time was longer than 2 minutes. Both groups used different laser systems (Catalys® and Victus®). Diakonis et al. compared three laser platforms (LenSx, Alcon Laboratories; Catalys, Abbott Medical Optics Inc.; and Victus, Bausch & Lomb, Inc.) and their effect on pupil diameter [12] and found a mean pupillary miosis of 1.42 ± 1.26 mm

for the LenSx, 0.66 ± 0.89 mm for the Catalys, and 0.14 ± 0.34 mm for the Victus group [7]. The difference between the three groups was statistically significant.

Although all three laser platforms use high-energy technology, their patient interface systems and other parameters are different. This emphasizes the importance of obtaining a better understanding of the different factors that affect miosis induction and prostaglandin release during FLACS. Interface systems are divided into applanation and nonapplanation types. The nonapplanation type induces minimal elevation of intraocular pressure (IOP; reportedly less than 40 mm Hg), whereas the applanation type, as used with the Ziemer 8 platform, induces an elevation of IOP of up to 80 mm Hg [8]. It is possible that prostaglandin release is initiated not only through direct shockwaves from laser pulses onto the ciliary body but also through unknown inflammatory processes upon release of the lens protein into the anterior chamber after completion of the capsulotomy or release of cytokines through the clear cornea incision or suction-induced IOP elevation. Schultz et al. found that $PGE_2$ increased after femtosecond laser capsulotomy alone but not after lens fragmentation without capsulotomy [28]. Jun [8] found that the anatomic distance between the laser capsulotomy and the pupil margin was significantly and positively correlated with pupil constriction, although this was contradicted by Popiela et al. [19], who reported sustained mydriasis in eyes with a mean capsulotomy–pupil distance of 0.99 ± 0.29 mm as opposed to decreasing pupil size in patients with a mean capsulotomy–pupil distance of only 0.77 ± 0.7 mm. Popiela et al. suggested that the gas bubbles created during capsulotomy irritated the pupil edge and led to prostaglandin release with subsequent constriction. Therefore, a larger distance protected the pupil margin from irritation through gas bubbles [5,19,26]. In the present study, the distance between the pupil diameter and capsulotomy during suction was greater than 1.5 mm in all cases, as the capsulotomy size was fixed at 5.2 mm. This might explain the observation of sustained mydriasis, despite the fact that no NSAIDs were administered before FLACS.

Of course, differences in the laser technology itself could also be a cause of the variations in pupil behavior during FLACS. "Low-energy" concepts with a high numerical aperture in femtosecond laser optics are an important evolutionary step toward smaller laser spots and thereby reduce collateral damage to the surrounding ocular tissue [29]. The modern low-energy concept combines high repetition rate (>1 MHz) and pulse energies in the nanojoule range, whereas "high-energy" femtosecond lasers emit energy in the microjoule range. Low-energy concepts aim to achieve precise tissue cuts while minimizing mechanical side effects [12,13]. The results of the present study can be explained by the fact that a low-energy laser platform probably produces lower collateral damage to the surrounding tissue, such as mechanical effects of cavitation bubbles, thus resulting in a smaller release of prostaglandins and, thereby, no intraoperative pupil narrowing. In fact, a very recent study on low-pulse energy femtosecond laser pretreatment did not note any additional interleukins but only a small, though statistically significant, increase of prostaglandin release in the anterior chamber compared with conventional phacoemulsification [21]. The researchers analyzed the level of inflammatory parameters 5 minutes after completion of laser pretreatment. A single dose of topical NSAID had been administered 30 minutes before the initiation of cataract surgery [21]. The authors concluded that the decrease in the inflammatory reaction as compared with values reported in the literature was a result of the lower pulse energy femtosecond laser. This finding is one possible explanation for the observation of an overall unchanged pupil area although the role of preoperative NSAID administration could have contributed to the low inflammatory cytokine levels observed by Schwarzenbacher et al Nevertheless, in a 2019 study by Liu et al., FLACS was performed using the same low-energy Femto LDV Z8 laser platform used in the present study, and a significantly higher $PGE_2$ level was induced as compared with conventional phacoemulsification [18]. The authors report clinically relevant pupil constriction. They observed

that preoperative administration of NSAIDS reduced the $PGE_2$ surge and occurrence of intraoperative miosis. Additionally, oxidative stress induced during phacoemulsification was measured by an increase in aqueous malondialdehyde and was strongly correlated with the effective phacoemulsification time but not with the femtosecond laser application or NSAID use. Although the patients in Schwarzenbacher et al.'s study were operated under topical anesthesia, patients in the current study and the research by Liu et al. were operated under local peribulbar anesthesia. Peribulbar anesthesia, when including the optic nerve, leads to pupillary dilation via an afferent pupillary defect in itself and blocks reactive miosis from the light of the operating microscope. A major difference between the above-mentioned studies and the current study is the fact that in the latter, no corneal cuts were made with the femtosecond laser. Creating a clear corneal incision with the femtosecond laser is an independent factor in FLACS-induced prostaglandin release [26].

Another possible explanation for the observation of an unchanged pupil area in the current study is the shifting time. When the Ziemer Z8 laser is used, surgery can be continued without any delay after completion of laser pretreatment. Other laser platforms require the patient to be transferred to another surgical table or theater. The increase in time elapsed allows the prostaglandins released after femtosecond laser pretreatment to have an effect on the pupil. This has been examined in different studies, and the results are inconclusive [30]. Schwarzenbacher et al. waited 5 minutes after completion of laser pretreatment and found FLACS-induced miosis [21], whereas Popiela et al. found FLACS-induced miosis but no correlation with their admittedly short shifting time of 4.22 minutes [19]. This was in accordance with the findings by Jun et al., who reported a shifting time of 40 minutes and no correlation with pupillary constriction [8].

The results of the present study are clinically relevant because a smaller pupil size is generally considered challenging for the surgeon and potentially leads to a higher incidence and/or severity of complications after cataract surgery [8,31]. It is, therefore, assumed that the absence of femtosecond laser-associated intraoperative miosis using a low-energy platform increases safety during surgery, makes surgery easier for the cataract surgeon, causes less trauma to the eye, and is therefore beneficial for patients.

The following limitations of this study merit consideration. (1) The mean difference plots show that the mean values were within the equivalence range, but not for every single measurement. Because the variance was comparable between preoperative and posttreatment data, it is assumed that the conclusion of equivalence of pupil sizes is indeed correct. A systematic effect (enlargement or reduction in size) would lead to changes in variance. (2) The study sample was entirely composed of Caucasian individuals. A different pupillary behavior could be suspected in more pigmented irides. (3) No intraoperative measurement of prostaglandins in aqueous humor was obtained. (4) There was a negligible shifting time. (5) One further limitation of the present study is that all study measurements were done under peribulbar anesthesia (as it is performed in 46% of the cataract surgeries in Germany; according to a survey in 2019 [32]). Further studies using other anesthesia methods are necessary to confirm the role of anesthetic method.

The results of the current study are valuable because they demonstrate that sustained mydriasis is possible in FLACS under specific surgical settings. Although further studies are needed, the goal should be to adjust these surgical settings to make FLACS an even more reliable and safe procedure for the benefit of patients.

In summary, this study demonstrates sustained mydriasis that even increased during suction when performing FLACs without preoperative use of NSAIDs. This contradicts the findings of various other studies but can be explained by the differences between the surgical settings of the present study and prior research, including the use of a low-energy, high-

frequency laser, omission of femtosecond laser corneal cuts, a large distance between the pupillary margin and capsulotomy, a negligible shifting time, and surgery performed under peribulbar anesthesia.

## Supporting information

**S1 Data.**
(XLSX)

## Acknowledgments

**Meeting presentation**: Presented in part at the ESCRS annual meeting, Lisbon, Portugal, October 2017.

## Author Contributions

**Conceptualization:** Alireza Mirshahi, Katharina A. Ponto.

**Data curation:** Alireza Mirshahi.

**Formal analysis:** Alireza Mirshahi, Astrid Schneider, Catharina Latz, Katharina A. Ponto.

**Investigation:** Alireza Mirshahi, Astrid Schneider.

**Methodology:** Alireza Mirshahi, Astrid Schneider, Katharina A. Ponto.

**Project administration:** Alireza Mirshahi.

**Resources:** Alireza Mirshahi.

**Software:** Astrid Schneider.

**Supervision:** Alireza Mirshahi, Katharina A. Ponto.

**Validation:** Alireza Mirshahi, Catharina Latz.

**Writing – original draft:** Alireza Mirshahi, Astrid Schneider, Catharina Latz, Katharina A. Ponto.

**Writing – review & editing:** Alireza Mirshahi, Astrid Schneider, Catharina Latz, Katharina A. Ponto.

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
