## [Decision Letter · Decision Letter 0]

10 Nov 2020

PONE-D-20-21038

Perioperative pupil size in low-energy femtosecond laser-assisted cataract surgery

PLOS ONE

Dear Dr. Mirshahi,

Thank you for submitting your manuscript to PLOS ONE. After careful consideration, we feel that it has merit but does not fully meet PLOS ONE’s publication criteria as it currently stands. Therefore, we invite you to submit a revised version of the manuscript that addresses the points raised during the review process.

We look forward to receiving your revised manuscript.

Kind regards,

Yu-Chi Liu, M.D

Academic Editor

PLOS ONE

Journal Requirements:

2.We note that you have indicated that data from this study are available upon request. PLOS only allows data to be available upon request if there are legal or ethical restrictions on sharing data publicly. For information on unacceptable data access restrictions, please see http://journals.plos.org/plosone/s/data-availability#loc-unacceptable-data-access-restrictions.

3.Thank you for stating the following in the Acknowledgments Section of your manuscript:

[K.A.P. is funded by the Federal Ministry of Education and Research (BMBF 01EO1003).]

 [AM is a consultant to Ziemer Ophthalmics, Biel, Switzerland

Yes: Ziemer assisted in extraction of pictures from the femtosecond laser device.]

Additionally, because some of your funding information pertains to commercial funding, we ask you to provide an updated Competing Interests statement, declaring all sources of commercial funding.

In your Competing Interests statement, please confirm that your commercial funding does not alter your adherence to PLOS ONE Editorial policies and criteria by including the following statement: "This does not alter our adherence to PLOS ONE policies on sharing data and materials.” as detailed online in our guide for authors  http://journals.plos.org/plosone/s/competing-interests.  If this statement is not true and your adherence to PLOS policies on sharing data and materials is altered, please explain how.

Please include the updated Competing Interests Statement and Funding Statement in your cover letter. We will change the online submission form on your behalf.

Reviewers' comments:

Reviewer's Responses to Questions

**Comments to the Author**

1. Is the manuscript technically sound, and do the data support the conclusions?

Reviewer #1: Yes

Reviewer #2: Yes

2. Has the statistical analysis been performed appropriately and rigorously? 

Reviewer #1: Yes

Reviewer #2: Yes

3. Have the authors made all data underlying the findings in their manuscript fully available?

Reviewer #1: Yes

Reviewer #2: Yes

4. Is the manuscript presented in an intelligible fashion and written in standard English?

Reviewer #1: Yes

Reviewer #2: Yes

5. Review Comments to the Author

Reviewer #1: Overall, this is a nice study with a thoughtful and well-researched discussion that contributes to our understanding of the effects of femtosecond laser technology for cataract surgery on pupillary dynamics. Most of my comments here pertain to style and grammatical suggestions.

Line 59-60: Do the authors mean that they generally did not observe pupillary miosis in their experience prior to conducting this study? The way in which it is worded makes it sound almost like a conclusion of the study rather than a reason to perform the study.

Line 149: Perhaps this can be phrased as “A secondary question was whether the values measured during different steps of the laser pretreatment varied from the initial values.”

Line 168: Should be “The mean (±SD), median, minimum[…]”

Line 176: “Change in pupil diameters at different time points{…}”

Line 197-200: Awkward phrasing and grammar.

Line 216: “in contrast to” may be a better phrase than “contradictory to”.

Line 217: I think it would help to specify that this was a study with low energy FLACS.

Line 222-224: Awkward phrasing.

Line 228-229: May be better to word this as “[…]and increased amounts have been measured in FLACS compared to classic phacoemulsification[…]”.

Line 231: “femtolaser” is nonstandard. Should say “femtosecond laser” or abbreviate all instances of “femtosecond laser” to something like “FSL”. This also occurs in Line 346.

Line 235: I suggest “NSAID pretreatment does not appear to completely eliminate FLACS-induced miosis.”

Line 245: Use of “femto” is nonstandard. See above.

Line 295: References to studies with multiple authors should be written with “et al”, as in “in a 2019 study by Liu et al.” This error occurs several times.

Lines 298-302: I believe all these statements are referring to the study by Liu et al., but it becomes less clear with each sentence. I suggest “They observed that preoperative administration of NSAIDS[…]” and “Additionally, oxidative stress induced[…]”, assuming this is what the authors mean.

Line 322-323: This seems like an orphaned sentence. Explanation and sources should be provided for this statement.

Other comments: I personally do not routinely administer peribulbar anesthesia for my cataract surgeries, preferring topical anesthesia especially with my FLACS cases. I wonder if the authors in their discussion can comment further on what extent they believe peribulbar anesthesia could have been a contributing factor in their observations if at all.

Reviewer #2: Line 41: Please reword FLACS has become Staple of modern cataract surgery. There is no evidence FLACS is superior to standard phaco with regards to any clinical outcomes

Line 168:

Please report this is a less confusing way

Mean (SD), median, and range

Line 115: Peribulbar anesthesia ? May be good to state exactly volume and needle used to administer this

Most cataract surgeries in U.S. are done under topical so this study would not be relevant. Authors should mention what percent of cases done with block in Germany If majority are not under topical, this study is not relevant given the block is likely the only reason constriction was not seen.

6. PLOS authors have the option to publish the peer review history of their article (what does this mean?). If published, this will include your full peer review and any attached files.

Reviewer #1: No

Reviewer #2: No

---

## [Author Response · Author response to Decision Letter 0]

13 Jan 2021

We have followed all suggestions by the two reviewers as follows:

Reviewer #1: Overall, this is a nice study with a thoughtful and well-researched discussion that contributes to our understanding of the effects of femtosecond laser technology for cataract surgery on pupillary dynamics. Most of my comments here pertain to style and grammatical suggestions.

1. Line 59-60: Do the authors mean that they generally did not observe pupillary miosis in their experience prior to conducting this study? The way in which it is worded makes it sound almost like a conclusion of the study rather than a reason to perform the study.

Thank you for this comment. We changed the sentence to “However, according to our personal experience in the clinic we hypothesized that no miosis induction through laser pretreatment occurs when using the Ziemer Z8 laser platform.”

2. Line 149: Perhaps this can be phrased as “A secondary question was whether the values measured during different steps of the laser pretreatment varied from the initial values.”

The sentence was changed as proposed.

3. Line 168: Should be “The mean (±SD), median, minimum[…]”

We changed this as proposed.

4. Line 176: “Change in pupil diameters at different time points{…}”

The sentence was corrected accordingly.

5. Line 197-200: Awkward phrasing and grammar.

The sentence was changed to “We could not detect any equivalence of the preoperative pupil diameters to those measured immediately after application of suction, after completion of lens fragmentation, or after completion of laser capsulotomy (suction applied during those measurements, TOST procedure, all p>0.05).”

6. Line 216: “in contrast to” may be a better phrase than “contradictory to”.

We changed it as proposed.

7. Line 217: I think it would help to specify that this was a study with low energy FLACS.

We added this.

8. Line 222-224: Awkward phrasing.

The sentence was changed to “This is a physiologic phenomenon which is explained both by the afferent pupillary defect and also by an impairment in the iris sphincter (due to an intraocular pressure elevation greater than the systolic ophthalmic artery pressure)”

9. Line 228-229: May be better to word this as “[…]and increased amounts have been measured in FLACS compared to classic phacoemulsification[…]”.

We changed the sentence as proposed.

10. Line 231: “femtolaser” is nonstandard. Should say “femtosecond laser” or abbreviate all instances of “femtosecond laser” to something like “FSL”. This also occurs in Line 346.

We replaced “femtolaser” by “femtosecond laser” throughout the manuscript.

11. Line 235: I suggest “NSAID pretreatment does not appear to completely eliminate FLACS-induced miosis.”

We changed the sentence as proposed.

12. Line 245: Use of “femto” is nonstandard. See above.

We changed this throughout the paper.

13. Line 295: References to studies with multiple authors should be written with “et al”, as in “in a 2019 study by Liu et al.” This error occurs several times.

We corrected this throughout the manuscript.

14. Lines 298-302: I believe all these statements are referring to the study by Liu et al., but it becomes less clear with each sentence. I suggest “They observed that preoperative administration of NSAIDS[…]” and “Additionally, oxidative stress induced[…]”, assuming this is what the authors mean.

The sentences were revised as proposed.

15. Line 322-323: This seems like an orphaned sentence. Explanation and sources should be provided for this statement.

The sentence referred to the sentence one line above. In fact, however, the wording was awkward. We therefore reworded the sentence to better reflect the reference to the study.

16. Other comments: I personally do not routinely administer peribulbar anesthesia for my cataract surgeries, preferring topical anesthesia especially with my FLACS cases. I wonder if the authors in their discussion can comment further on what extent they believe peribulbar anesthesia could have been a contributing factor in their observations if at all.

We added the following paragraph to the discussion: “One further limitation of the present study is that all study measurements were done under peribulbar anesthesia (as it is performed in 46% of the cataract surgeries in Germany; according to a survey of 2019 by the German Society of Intraocular Implants and Refractive Surgery, DGII […]). We do not believe that the results would have been different under other anesthesia methods (incl. topical); However, further studies using other anesthesia methods are necessary to confirm the assumed.” (lines 338-344)

17. Reviewer #2: Line 41: Please reword FLACS has become Staple of modern cataract surgery. There is no evidence FLACS is superior to standard phaco with regards to any clinical outcomes

We changed the sentence to: “During the past decade, femtosecond laser-assisted cataract surgery (FLACS) has gained increasing popularity”

 

18. Line 168: Please report this is a less confusing way; Mean (SD), median, and range

This has been changed (see #3).

19. Line 115: Peribulbar anesthesia ? May be good to state exactly volume and needle used to administer this.

We added the details (ieach 3 ml of Mepivacaine 2% and Bupivacaine 0.75% and 75 IE hyaluronidase with a 22 Gauge needle). Lines 116-118

20. Most cataract surgeries in U.S. are done under topical so this study would not be relevant. Authors should mention what percent of cases done with block in Germany If majority are not under topical, this study is not relevant given the block is likely the only reason constriction was not seen.

We added the following paragraph to the discussion: “One further limitation of the present study is that all study measurements were done under peribulbar anesthesia (as it is performed in 46% of the cataract surgeries in Germany; according to a survey of 2019 by the German Society of Intraocular Implants and Refractive Surgery, DGII […]). We do not believe that the results would have been different under other anesthesia methods (incl. topical); However, further studies using other anesthesia methods are necessary to confirm the assumed.” (lines 338-344)

---

## [Decision Letter · Decision Letter 1]

16 Feb 2021

PONE-D-20-21038R1

Perioperative pupil size in low-energy femtosecond laser-assisted cataract surgery

PLOS ONE

Dear Dr. Mirshahi,

Thank you for submitting your manuscript to PLOS ONE. The reviewers have raised some more comments. Therefore, we invite you to submit a revised version of the manuscript that addresses the points raised during the review process.

We look forward to receiving your revised manuscript.

Kind regards,

Yu-Chi Liu, MD, MCI

Academic Editor

PLOS ONE

Reviewers' comments:

Reviewer's Responses to Questions

**Comments to the Author**

1. If the authors have adequately addressed your comments raised in a previous round of review and you feel that this manuscript is now acceptable for publication, you may indicate that here to bypass the “Comments to the Author” section, enter your conflict of interest statement in the “Confidential to Editor” section, and submit your "Accept" recommendation.

Reviewer #1: All comments have been addressed

Reviewer #2: All comments have been addressed

2. Is the manuscript technically sound, and do the data support the conclusions?

Reviewer #1: Yes

Reviewer #2: No

3. Has the statistical analysis been performed appropriately and rigorously? 

Reviewer #1: Yes

Reviewer #2: Yes

4. Have the authors made all data underlying the findings in their manuscript fully available?

Reviewer #1: Yes

Reviewer #2: Yes

5. Is the manuscript presented in an intelligible fashion and written in standard English?

Reviewer #1: Yes

Reviewer #2: Yes

6. Review Comments to the Author

Reviewer #1: Additional minor recommendations and comments:

Line 253: “Although all three laser platforms […]” can be the start of a new paragraph.

Line 293: If subjects were given a dose of NSAID before surgery, I’m not sure how it can be concluded that the decrease in inflammatory reaction results from the use of lower pulse energy FSL and not from the use of pre-operative NSAIDs. Authors should confirm whether this was in fact a conclusion of the cited study and/or clarify this statement.

Line 323-324: This is still an orphaned sentence and does not appear to have been modified from the original. I’m uncertain as to its relevance, but it can probably be incorporated into another paragraph earlier in the manuscript if you choose to keep it.

Line 342: There is no reason to abbreviate “including”. The semicolon should be a period.

Lines 339-343: I would advise caution with this broad assumption as it seems to directly contradict your statement on the pupillary effects of peribulbar anesthesia in Lines 306-308.

Reviewer #2: It is not accurate to say peribulbar anesthesia does not affect pupil size. This is a fatal flaw of the manuscript. If 46% of cases in Germany done under peribulbar, then you should review the results of topical as well. Not sure why those with topical anesthesia were excluded from the study.

https://pubmed.ncbi.nlm.nih.gov/23879851/

https://bjo.bmj.com/content/bjophthalmol/78/1/41.full.pdf

7. PLOS authors have the option to publish the peer review history of their article (what does this mean?). If published, this will include your full peer review and any attached files.

Reviewer #1: No

Reviewer #2: No

---

## [Author Response · Author response to Decision Letter 1]

22 Mar 2021

Response to Reviewers

Reviewer #1: Additional minor recommendations and comments:

Line 253: “Although all three laser platforms […]” can be the start of

a new paragraph.

 A new paragraph was included as recommended.

Line 293: If subjects were given a dose of NSAID before surgery, I’m

not sure how it can be concluded that the decrease in inflammatory

reaction results from the use of lower pulse energy FSL and not from

the use of pre-operative NSAIDs. Authors should confirm whether this

was in fact a conclusion of the cited study and/or clarify this

statement.

 As stated in the section “surgical technique” no NSAID was given to any patient before surgery in our study.

 With regards to Schwarzenbacher’s paper we acknowledge the anti-inflammatory effect of preoperative NSAID; We believe the authors’ conclusion is based on the fact that the measured inflammatory cytokine levels were lower than reported in other studies using other laser platforms. Our wording is exactly the conclusion of the cited study. Nevertheless, we added this issue to the discussion (Line 296-297)

Line 323-324: This is still an orphaned sentence and does not appear

to have been modified from the original. I’m uncertain as to its

relevance, but it can probably be incorporated into another paragraph

earlier in the manuscript if you choose to keep it.

 The sentence was deleted.

Line 342: There is no reason to abbreviate “including”. The semicolon

should be a period.

The changes were made as recommended.

Lines 339-343: I would advise caution with this broad assumption as it

seems to directly contradict your statement on the pupillary effects

of peribulbar anesthesia in Lines 306-308.

 We revised the wording and hope it is more suitable now. Lines 342-346

Reviewer #2: 

It is not accurate to say peribulbar anesthesia does not

affect pupil size. This is a fatal flaw of the manuscript. If 46% of

cases in Germany done under peribulbar, then you should review the

results of topical as well. Not sure why those with topical anesthesia

were excluded from the study.

The authors do not claim peribulbar anesthesia does not affect the pupil size. We just believe the results of our study would have been similar under other anesthetic methods. Nevertheless, we revised the wording and hope it is more suitable (Lines 342-346). The authors perform almost all cataract surgeries under peribulbar anesthesia, thus the paper is solely representing those cases.

---

## [Decision Letter · Decision Letter 2]

29 Apr 2021

Perioperative pupil size in low-energy femtosecond laser-assisted cataract surgery

PONE-D-20-21038R2

Dear Dr. Mirshahi,

We’re pleased to inform you that your manuscript has been judged scientifically suitable for publication and will be formally accepted for publication once it meets all outstanding technical requirements.

Kind regards,

Yu-Chi Liu, M.D

Academic Editor

PLOS ONE

Additional Editor Comments (optional):

Reviewers' comments:

Reviewer's Responses to Questions

**Comments to the Author**

1. If the authors have adequately addressed your comments raised in a previous round of review and you feel that this manuscript is now acceptable for publication, you may indicate that here to bypass the “Comments to the Author” section, enter your conflict of interest statement in the “Confidential to Editor” section, and submit your "Accept" recommendation.

Reviewer #2: All comments have been addressed

2. Is the manuscript technically sound, and do the data support the conclusions?

Reviewer #2: Yes

3. Has the statistical analysis been performed appropriately and rigorously? 

Reviewer #2: Yes

4. Have the authors made all data underlying the findings in their manuscript fully available?

Reviewer #2: Yes

5. Is the manuscript presented in an intelligible fashion and written in standard English?

Reviewer #2: Yes

6. Review Comments to the Author

Reviewer #2: Comments addressed.

I have stated the limitations. In the U.S. most cataract cases are performed under topical, so I think studying this in topical cases would be more helpful.

7. PLOS authors have the option to publish the peer review history of their article (what does this mean?). If published, this will include your full peer review and any attached files.

Reviewer #2: No

---

## [Editor Report · Acceptance letter]

7 May 2021

PONE-D-20-21038R2 

Perioperative pupil size in low-energy femtosecond laser-assisted cataract surgery 

Dear Dr. Mirshahi:

I'm pleased to inform you that your manuscript has been deemed suitable for publication in PLOS ONE. Congratulations! Your manuscript is now with our production department. 

Kind regards, 

on behalf of

Dr. Yu-Chi Liu 

Academic Editor

PLOS ONE